# Examining the impact of ICU population interaction structure on modeled colonization dynamics of *Staphylococcus aureus*

**Matthew S. Mietchen**[1], **Christopher T. Short**[1], **Matthew Samore**[2,3], **Eric T. Lofgren**[1]*, **CDC Modeling Infectious Diseases in Healthcare Program (MInD-Healthcare)**¶

**1** Paul G. Allen School for Global Health, College of Veterinary Medicine, Washington State University, Pullman, Washington, United States of America, **2** Department of Internal Medicine, University of Utah School of Medicine, University of Utah, Salt Lake City, Utah, United States of America, **3** VA Salt Lake City Healthcare System, Salt Lake City, Utah

¶ Membership of MInD-Healthcare is provided in the Acknowledgments.
* Eric.Lofgren@wsu.edu

**Data Availability Statement:** The code and data for the model simulation and subsequent analysis

## Abstract

### Background

Complex transmission models of healthcare-associated infections provide insight for hospital epidemiology and infection control efforts, but they are difficult to implement and come at high computational costs. Structuring more simplified models to incorporate the heterogeneity of the intensive care unit (ICU) patient-provider interactions, we explore how methicillin-resistant Staphylococcus aureus (MRSA) dynamics and acquisitions may be better represented and approximated.

### Methods

Using a stochastic compartmental model of an 18-bed ICU, we compared the rates of MRSA acquisition across three ICU population interaction structures: a model with nurses and physicians as a single staff type (SST), a model with separate staff types for nurses and physicians (Nurse-MD model), and a Metapopulation model where each nurse was assigned a group of patients. The proportion of time spent with the assigned patient group (γ) within the Metapopulation model was also varied.

### Results

The SST, Nurse-MD, and Metapopulation models had a mean of 40.6, 32.2 and 19.6 annual MRSA acquisitions respectively. All models were sensitive to the same parameters in the same direction, although the Metapopulation model was less sensitive. The number of acquisitions varied non-linearly by values of γ, with values below 0.40 resembling the Nurse-MD model, while values above that converged toward the Metapopulation structure.

### Discussion

Inclusion of complex population interactions within a modeled hospital ICU has considerable impact on model results, with the SST model having more than double the acquisition rate of

may be found at github.com/epimodels/
Metapopulation_MRSA.

**Funding:** This work was supported by the CDC
Cooperative Agreement RFA-CK-17-001-Modeling
Infectious Diseases in Healthcare Program (MInD-
Healthcare). As part of the MInD-Healthcare
program, the funder had input on the design of the
study as well as an opportunity to review the
manuscript prior to submission.

**Competing interests:** The authors have declared
that no competing interests exist.

the more structured metapopulation model. While the direction of parameter sensitivity remained the same, the magnitude of these differences varied, producing different colonization rates across relatively similar populations. The non-linearity of the model's response to differing values of a parameter gamma ($\gamma$) suggests simple model approximations are appropriate in only a narrow space of relatively dispersed nursing assignments.

## Conclusion

Simplifying assumptions around how a hospital population is modeled, especially assuming random mixing, may overestimate infection rates and the impact of interventions. In many, if not most, cases more complex models that represent population mixing with higher granularity are justified.

### Author summary

Some models of healthcare-associated infection assume random mixing between healthcare workers and patients–that is, all healthcare workers care for all patients in the model at equal frequency. However, in many settings, healthcare workers are not uniformly distributed among patients due to scheduling, patient complexity or cohorting, the built environment, or hospital policy. Nevertheless, models that assume random mixing are often chosen for analytical tractability or computational speed. This paper explores the impact of assuming random mixing by comparing a model of an 18-bed intensive care unit that assumes random mixing with one that assigns a group of patients to each nurse while a dedicated critical care physician continues to see all patients. Higher rates of segmentation result in lower rates of predicted acquisitions of methicillin-resistant *Staphylococcus aureus*, as well as lower predicted impacts of interventions. These findings suggest that the simpler random mixing models may not be appropriate approximations in settings where healthcare workers are not uniformly distributed among patients, and that balancing the computational costs with the trend toward more complex models in hospital epidemiology is justified.

## Introduction

Dynamic transmission models have provided valuable insight toward controlling healthcare-associated infections (HAIs) for decades, particularly in addressing intervention effectiveness to limit colonization and spread of pathogens between hospitalized patients and healthcare workers [1–5]. The complexity and variation of HAI transmission models, as well as the advancement of methods for model fitting and sensitivity analysis, has increased over time. Methods used in HAI modeling studies vary widely, from adaptations of the classic Ross-McDonald model [6] that are simplified but analytically approachable [7–11] to complex network or agent-based models that include high fidelity representations of patient-to-staff interactions but are correspondingly more complex and vulnerable to results arising from subtle and unintuitive interactions between agents [12–17]. However, a 2013 systematic review by van Kleef *et al.* found that most HAI modeling studies used homogenous mixing compartmental models with limited hospital structure [18].

Concerns regarding the heterogeneous contact patterns that exist in hospitals suggest the need for network and agent-based models. The use of agent-based models for HAI modeling studies has increased greatly in recent years. A systematic review performed by Nguyen *et al.* in 2019 noted that while "systems dynamics models"–their term for compartmental models with homogeneous mixing–accounted for 38% of the models they reviewed, agent-based and discrete event models already accounted for the same volume of studies (38%), despite beginning to appear almost a decade later. While the authors note agent-based models can overcome the limitations of systems dynamics models, they also highlight the high computational costs, need for extensive data, and accompanying uncertainty analysis as major limitations [19]. These limitations do affect the generalizability and reproducibility of agent-based modeling studies, but efforts to address and mitigate this effect are underway [20].

Infectious disease research continues to grapple with the question of what level of complexity is required for robust results [21,22]. Complex network or agent-based models are computationally intensive and require a degree of software engineering expertise, whereas simpler models are more accessible. Rarely, however, are the sensitivity of the resulting models to structure decisions examined. If, and under what circumstances, simpler model formulations may be acceptable approximations of more complicated models, and what impacts these simplifications have on the model findings, remains an area that is underdeveloped within healthcare-associated infection modeling.

As an approach to understanding and quantifying these trade-offs, we consider the impact of structured contacts within a compartmental modeling framework using Methicillin-resistant *Staphylococcus aureus* (MRSA) as a motivating example. MRSA is a well-studied infection that is important in intensive care settings [2,3,10,19,23–28]. The vulnerable nature of the patients and the difficulty in treating severe infections makes preventing the spread of this antibiotic-resistant pathogen a priority among infection control efforts.

Patients admitted to the ICU have been found to have persistent colonization with MRSA 12–14 days after discharge from the hospital [29]. There is strong evidence that frequency and patterns of interaction between staff and patients play a critical role in transmission [30]. The ratio of nurses to patients has been found to contribute to the overall level of pathogen colonization and transmission within healthcare settings and ICUs [31–34]. Higher patient to nurse ratios have a positive correlation with increased transmission and poor health outcomes [34]. Hospital staffing levels, patient interactions, surveillance detection, and important parameters such as hand hygiene have been explored for several decades using mathematical models [2,3,25,26].

Using a stochastic compartmental model of an 18-bed ICU, we compared three potential population interaction structures: a single-staff-type model where all healthcare workers interact randomly with all patients, and there is no differentiation between nurses and physicians; a model that divides healthcare workers into nurses and physicians but continues to assume random mixing; and a highly structured model where each nurse is assigned a specific group of patients, i.e. a metapopulation-like structure. These models are hereafter referred to as "SST", "Nurse-MD", and "Metapopulation", respectively.

We also developed hybrid model, which reflects the limited random interaction resulting from variance from patient assignments often seen in the ICU environment, such as cross-coverage during breaks, staff shortages, or complex procedures that require higher numbers of healthcare workers. For some portion of the work day, a nurse may to randomly interact with patients not originally under their direct care. The model accounts for this variation by utilizing a metapopulation that is primarily but not exclusively, organized into distinct subpopulations but still allows for some interaction between nurses and all patients.

These models represent moving from an extremely simple model structure to one that is of intermediate complexity, and finally to a model that approximates same interactions that might appear in a network or agent-based model while still adhering to the compartmental model framework (allowing it to use the same parameters and for differences in software implementation to be ignored in understanding the differences between models). In doing so, we examined the entire staff-patient interaction spectrum from random mixing to a highly structured model. We explored the sensitivity of these different models to changes in their underlying parameters, as reflected by the predicted number of MRSA acquisitions.

## Methods

### Model structure

MRSA transmission was simulated in an 18-bed medical ICU that included six nurses and a dedicated critical care physician based on a previously published model [27]. Three models were implemented with varying structures of provider-patient interactions. While ICU size and staffing levels can vary, we chose an 18-bed ICU with a patient-to-nurse ratio of 3:1 based on averages obtained from a large multicenter clinical trial [35]. A summary of all transmission model parameters can be found in Table 1.

In the single staff type (SST) model, patients are assumed to mix randomly with healthcare workers (HCWs), with no distinction between the nurses or the physician (Fig 1A). Hospital staff are either uncontaminated ($S_U$) or contaminated ($S_C$), representing infectious material on their hands or person. Patients are either uncolonized ($P_U$) or colonized ($P_C$). This model,

**Table 1. Parameters for modeling the acquisition of methicillin-resistant *Staphylococcus aureus* in an Intensive Care Unit.**

| Parameter | Parameter Description | Parameter Value | Source |
|---|---|---|---|
| $\rho$ | Contact rate between patients and HCWs | 4.154 (# of direct care tasks/hour) | [7,37] |
| $\rho_N$ | Contact rate between patients and nurses | 3.973 (# of nurse direct care tasks/hour) | [7,37] |
| $\rho_D$ | Contact rate between patients and physician | 0.181 (# of physician direct care tasks/hour) | [7,37] |
| $\sigma$ | Probability that a HCW's hands are contaminated from a single contact with a colonized patient | 0.054 | [38] |
| $\psi_{SST}$ | Probability of successful colonization of an uncolonized patient due to contact with a contaminated HCW when randomly mixed | 0.1494 | Fit to [35] |
| $\psi_{Nurse-MD}$ | Probability of successful colonization of an uncolonized patient due to contact with a contaminated HCW with physician separated | 0.1660 | Fit to [35] |
| $\psi_{Metapopulation}$ | Probability of successful colonization of an uncolonized patient due to contact with a contaminated HCW in metapopulation structure | 0.4481 | Fit to [35] |
| $\theta$ | Probability of discharge | 4.39 days$^{-1}$ | [35] |
| $\nu_u$ | Proportion of admissions uncolonized with MRSA | 0.9221 | [35] |
| $\nu_c$ | Proportion of admissions colonized with MRSA | 0.0779 | [35] |
| $\iota$ | Effective hand-decontaminations/hour (direct care tasks × hand hygiene compliance × efficacy) | 5.740 (10.682 direct care tasks/hour × 56.55% compliance × ~ 95% efficacy) | [7,35,37,39] |
| $\iota_N$ | Effective nurse hand-decontaminations/hour | 6.404 (11.92 direct care tasks/hour × 56.55% compliance × ~ 95% efficacy) | [7,35,37,39] |
| $\iota_D$ | Effective physician hand-decontaminations/hour | 1.748 (3.253 direct care tasks/hour × 56.55% compliance × ~ 95% efficacy) | [7,35,37,39] |
| $\tau$ | Effective gown or glove changes/hour (2 × # of visits × compliance) | 2.445 (2.957 changes/hour × 82.66% compliance) | [35,38,40] |
| $\tau_N$ | Effective nurse gown or glove changes/hour | 2.728 (3.30 changes/hour × 82.66% compliance) | [35,38,40] |
| $\tau_D$ | Effective physician gown or glove changes/hour | 0.744 (0.90 changes/hour × 82.66% compliance) | [35,38,40] |
| $\mu$ | Natural decolonization rate | 20.0 days$^{-1}$ | [41] |
| $\gamma$ | Proportion of time nurses spend with assigned patients | Varied between 1/6 and 1 | |

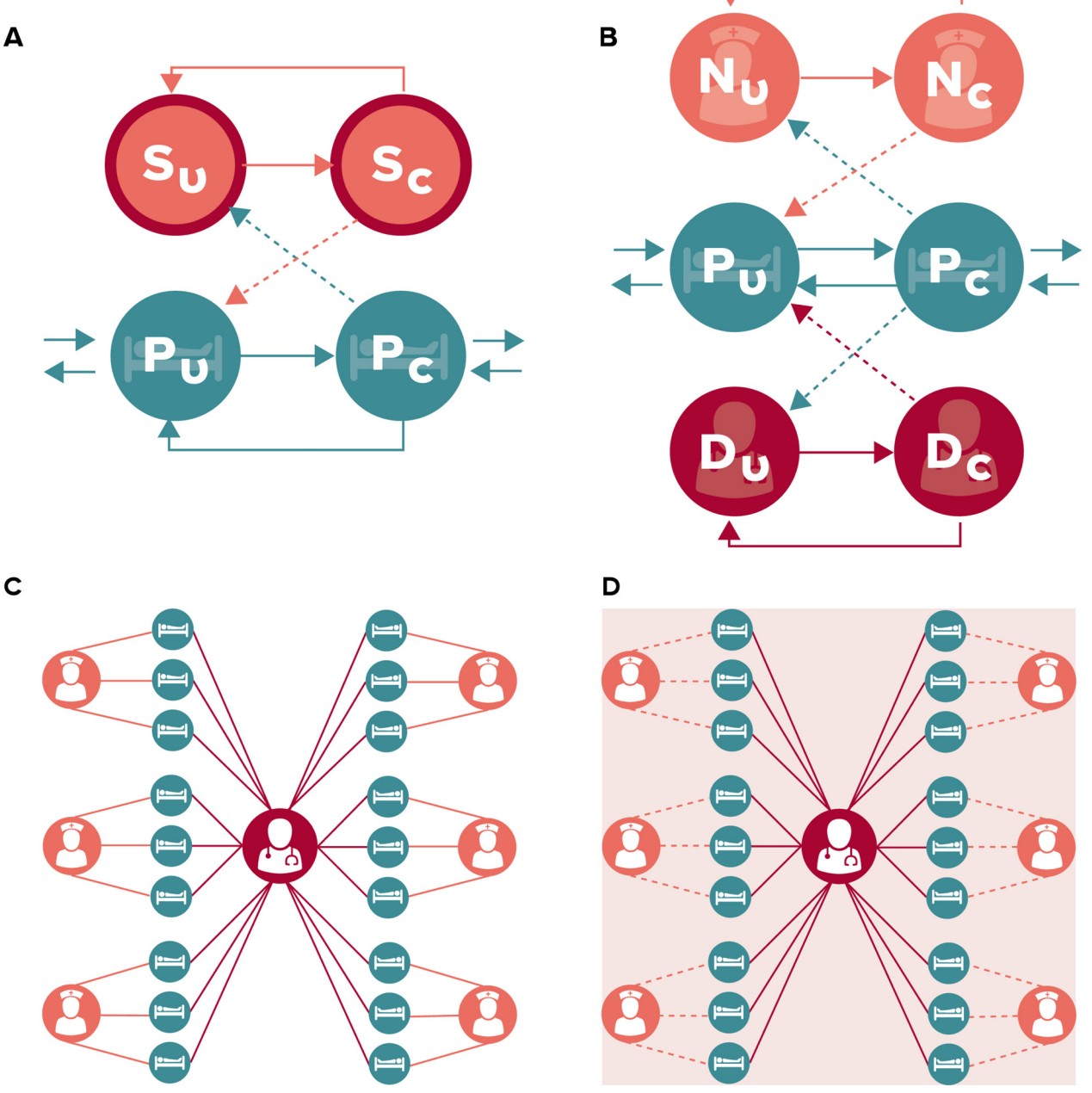

**Fig 1. Compartment models of methicillin-resistant Staphylococcus aureus (MRSA) acquisitions.** Patients and hospital staff are classified as (un)colonized or (un)contaminated (U and C on diagrams), respectively. Solid arrows indicate transition states, while dashed arrows indicate routes of MRSA transmission (transition parameters and equations are in Tables 1 and S1, respectively. A) Single Staff Type model, B) Nurse-MD model, C) Metapopulation model, D) a hybrid model where nurses only spend a fraction of their time in their assigned patient groups and otherwise see patients at random.

while unrealistic, was important to use as a baseline for comparison as it represents a homogenous mixing population often found in compartmental models. This simplified representation is also similar to how some agent-based models of larger-scale hospital networks represent single hospitals [13]. The SST model equations are available in S1 Table.

The "Nurse-MD" model retained random mixing with all patients but separated the physician from the nursing staff as two distinct populations (Fig 1B). Separation of the physician also allowed the interactions between healthcare workers and the patients to be more realistic, using role-specific contact rates with patients. In this model, physicians have less direct care tasks (touching the patient or their immediate surrounding environment) when compared to either nurses or the generic healthcare workers in the SST model. This model had six compartments within it: the number of patients either colonized ($P_C$) or uncolonized ($P_U$), the number of nurses either contaminated ($N_C$) or uncontaminated ($N_U$), and the two additional compartments representing the physician as either contaminated ($D_C$) or uncontaminated ($D_U$). The equations for the Nurse-MD model may be found in S2 Table.

The "Metapopulation model" further segregated the healthcare workers by assigning each nurse a specific group of patients (one nurse for every three patients) and assuming the nurse cared exclusively for those patients. This practice is common in many ICUs for continuity of care, familiarity, and scheduling purposes. The model compartments thus become further divided into six subpopulations, with the physician acting as a bridge between them (Fig 1C). This model structure creates a metapopulation that better represents an actual ICU organizational staffing structure at the cost of increased complexity and reduced analytical tractability. This model only assumes that the nurse visits each patient in their assigned group randomly. The equations for the Metapopulation model may be found in S3 Table.

Finally, we created a Hybrid model to explore intermediate population interactions between purely random mixing and strict nurse-patient groupings (Fig 1D). In this model, nurses were assigned to a specific group of patients but also interact with patients outside their assigned group due to cross-coverage, staff breaks, or patient care tasks that require more than one nurse to perform. This model adds a parameter, gamma (γ), to represent the amount of time a nurse spends in their assigned group, with the remainder of the time spent moving randomly among patients outside their assignment. When γ = 1/6, this model replicates the Nurse-MD model, as a nurse is no more likely to spend time with their assigned patients as they are any other five patient groups. Similarly, when γ = 1, the model replicates the Metapopulation model, where nurses only treat their assigned patients.

Several assumptions underlie all four models. First, patients are assumed to have a single-occupancy room, do not leave their room at any time, and therefore do not interact with other patients. It is assumed that nurses and the physician only interact with the patients and do not interact with each other in ways relevant to pathogen transmission. The ICU is considered a "closed ICU", meaning physicians or other hospital staff from outside the ICU do not interact with patients. The ICU is also considered to be at 100% capacity at all times, therefore if a patient is discharged it is assumed another patient is admitted to the bed immediately [36]. A hand hygiene opportunity occurs after every direct care task or any contact between a healthcare provider and a patient. Personal protective equipment (PPE) such as gowns and gloves are changed on entry and exit from the rooms of all colonized patients. Both hand hygiene and PPE are performed with imperfect compliance (Table 1). Lastly, we assumed that MRSA colonization is detected instantly and with perfect sensitivity and specificity to simplify the model, and that no treatments or interventions were performed for colonized patients other than the natural decolonization parameter, mu (μ).

## Parameterization

Parameter values were obtained predominantly from a previously published model of MRSA transmission in an ICU [35] and are described in Table 1. The Nurse-MD and Metapopulation models introduce new interactions between the patient and their healthcare team, which

required rederivation of some parameters from their original sources [40,42,43]. Specifically, the hand hygiene and gown/glove change rates incorporate nurse and physician specific contact rates, which were recalculated using the same methods as in the previous work.

Contact rates between patients and healthcare workers were represented by direct care tasks per hour for each healthcare worker type. Direct care tasks are defined as the physical interaction of the healthcare worker with the patient or their surrounding environment [40]. Effective hand-decontaminations per hour ($\iota$) were calculated by the number of direct care tasks and taking into consideration the compliance rate and handwashing efficacy. Effective gown and glove changes per hour ($\tau$) were calculated based on the number of visits to a patient per hour and a compliance rate–changing gowns and gloves was assumed to be 100% effective at removing contamination from a healthcare worker.

One additional parameter was added to the model differing from previously published work. A natural decolonization rate based on results from the STAR*ICU Trial was added based on evidence that colonization of MRSA is limited, and natural decolonization can occur without targeted treatment or decontamination efforts, moving patients from $P_C$ to $P_U$ at a low rate absent any direct intervention [29,41].

## Model simulation

The SST, Nurse-MD and Metapopulation models were simulated to count the number of patients who transitioned to the colonized state ($P_C$) in order to compare the average number of MRSA acquisitions. The models were stochastically simulated using Gillespie's Direct Method [44] in Python 3.6 using the StochPy package [45] for 1,000 iterations per model. The initial conditions for each model were set to have no contaminated healthcare workers and no colonized patients, with initial MRSA infections being seeded from colonized members of the community being admitted to the ICU. Each iteration was run for a single year. The distribution of the acquisitions for each model's 1,000 iterations was visualized in R v3.5.1 using the vioplot package [46], and the difference between them assessed using a Kruskal-Wallis test. The code for the model simulation and subsequent analysis may be found at github.com/epi-models/Metapopulation_MRSA.

## Model recalibration

In addition to considering model outcomes using a single set of parameters (originally calibrated to the SST model), we also examined the difference in the estimated value of a single free parameter which could be fit within each model. The purpose of this recalibration is two-fold. First, it allows for a comparison of the models in a setting where their outcomes are equal. Second, it allows us to examine how each model form might influence the value of an estimated parameter–important information in a setting where models may be used to perform statistical inference and estimate intervention efficacy. The parameter chosen for this recalibration, $\psi$, is the probability of an effective colonization of an uncolonized patient from contact wtih a contaminated healthcare worker.

Approximate Bayesian Computation (ABC) [47] was used for the parameter fitting and to obtain an approximate Bayesian posterior of $\psi$ for the SST, Nurse-MD, and Metapopulation models. This method samples a candidate value from a prior distribution, performs the model simulation using that candidate, and compares a summary statistic from that simulation to a target statistic. The candidate value is accepted if the simulation's summary statistic equals the target statistic $\pm$ an error term episilon ($\varepsilon$). This is performed repeatedly, and the resulting distribution of accepted candidates approximates a Bayesian posterior distribution.

For this analysis, the target number of acquisitions was set to 5.94 acquisitions per 1,000 person-days with an episilon ($\varepsilon$) of 15%, matching the rate seen in the control arm of a large randomized clinical trial on MRSA prevention during the study period [35]. A uniform prior bounded by 0.0 and 1.0 was used, and 1,000,000 candidate parameters were drawn from this distribution to obtain the approximated Bayesian posterior of $\psi$ for each model, using a simulation procedure similar to the one described above. For comparison between models, the median of this distribution was used as the value for $\psi$.

## Parameter sensitivity analysis

In addition to assessing the difference in raw acquisitions in each model, we assessed the sensitivity of this outcome to changes in the model's parameters. All parameters in the model were allowed to vary uniformly ±50% of their original values, and 100,000 parameter combinations were simulated for each model. For each model, the recalibrated value for $\psi$ was used in order to ensure the models were compared against a consistent acquisition rate. The number of acquisitions in each simulation was then normalized as a percentage-change from the mean number of acquisitions. Linear regression was used on the normalized acquisition rate to determine the percentage change in acquisitions due to a single-percentage change in each parameter value.

The Hybrid model was used to explore a more structural sensitivity question within the Metapopulation model by varying the amount of time a nurse spends exclusively with their assigned group vs. other patients on the ward, $\gamma$. The Hybrid model was simulated 10,000 times, drawing a value of $\gamma$ for each iteration from a uniform distribution bounded by 1/6 and 1. A segmented Poisson regression model was then fit to detect any thresholds in the value of $\gamma$ where it's relationship to the rate of MRSA acquisitions notably changed, or if the transition between the Nurse-MD model ($\gamma = 1/6$) and the Metapopulation model ($\gamma = 1$) was linear. This model incorporated linear and quadratic terms for $\gamma$ and allowed the model to choose any number of break points.

## Results

### Model comparison

When using the same parameter set (calibrated to the SST model), the probability density and average number of MRSA acquisition were significantly different between the SST, Nurse-MD and Metapopulation models ($\chi^2 = 1786.5$, df = 2, p > 0.001) (Fig 2). Using the SST model as the baseline for comparison, a decrease in the average number of MRSA acquisitions were observed in both the separate Nurse-MD model and the Metapopulation model. By separating the physician from the nurses, the mean acquisitions decreased 20.6% from 40.7 acquisitions to 32.3 acquisitions, respectively. Limiting the nurses' interaction to an assigned patient group yielded mean acquisitions of 19.8, a 51.4% decrease as compared to the original SST model.

### Model recalibration

The model parameter $\psi$ is the probability of effective colonization of an uncolonized patient from contact with a contaminated healthcare worker and was used to calibrate each of the models. Calibration of the SST model resulted in the median value of the parameter of 0.024 (95% Credible Interval: 0.016, 0.034). The Nurse-MD model results were very similar to the SST model, with a median value of 0.029 (95% Credible Interval: 0.019,0.042). In contrast, the Metapopulation model had a median $\psi$ value of 0.046 (95% Credible Interval: 0.032, 0.07), both a substantially higher estimate than the other models and one in which the bounds of the

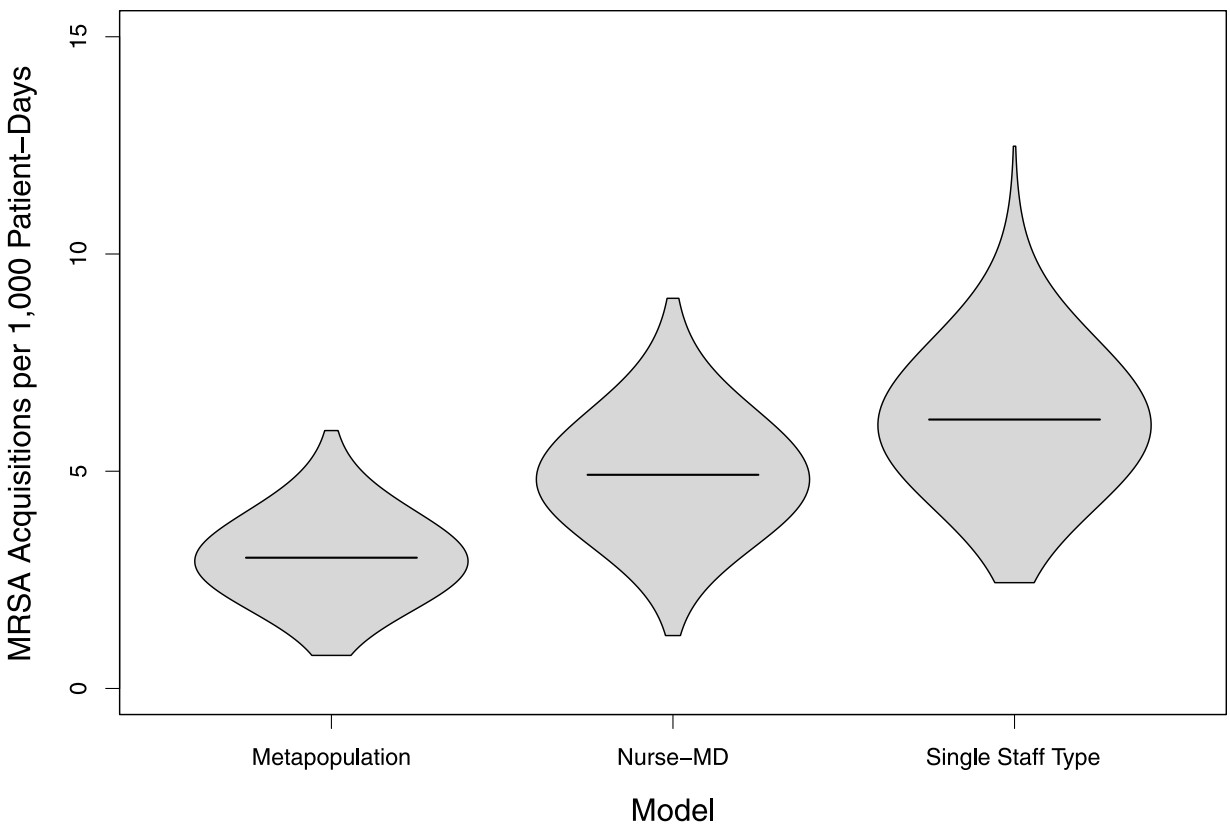

**Fig 2. Distribution of cumulative MRSA acquisitions in 3,000 simulated 18-bed intensive care units under three theoretical population structures.**

credible interval did not contain the other estimates. The altered contact patterns in the Metapopulation model require substantially higher per-contact colonization probabilities to sustain the same level of colonization.

## Sensitivity analysis

While the Metapopulation model resulted in fewer acquisitions, certain parameters were found to affect the model outcomes to a larger magnitude when compared to the other models. The three parameters showing the largest proportional change ($> 0.20$) in cumulative acquisitions (Fig 3A) were contact rate ($\rho$), probability of patient colonization ($\psi$), and hand-decontamination ($\iota$). We made similar findings for the Nurse-MD model, though generally only for the nurse-specific parameters (Fig 3B). The doctor-specific parameters had little effect on the model outcomes. Only one parameter of the Metapopulation model had a large change in cumulative acquisitions ($> 0.20$)–the nurse-specific contact rate ($\rho_N$). This is consistent with the previous two models (Fig 3C), although the effect was attenuated.

The directionality of the overall change in cumulative acquisitions by parameter is an important measure of model stability and correct parameter estimates, as this reflects whether the models *qualitatively* give the same results as to whether or not a particular parameter value changing results in an increase or decrease in MRSA acquisitions, even if the models disagree as to the specific value of that change. All the parameters between the models are consistent in terms of directionality, with the Metapopulation model having a smaller change in magnitude of the cumulative acquisitions (Fig 3D).

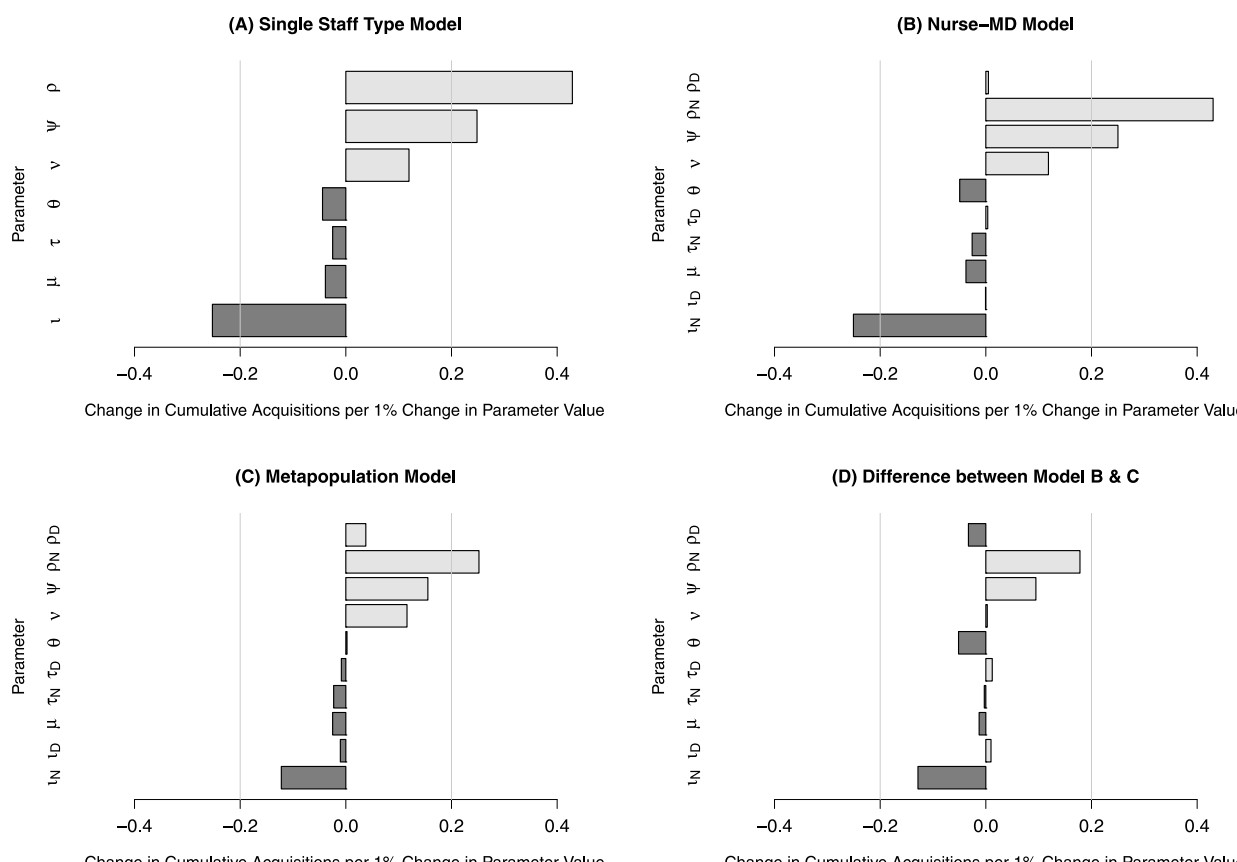

**Fig 3. Global parameter sensitivity of three modeled ICU population structures.** Panel A depicts the change in proportional change in cumulative MRSA acquisitions per one-percent change in the value of a specific parameter, with light bars indicating increased acquisitions, and dark bars indicating decreased acquisitions for a model assuming random mixing and with a single staff type for both nurses and physicians. Pale grey vertical lines indicate a change greater than 0.2 in either direction, which was used as a boundary condition for major changes. Panel B depicts the same for a model that separates nurses and physicians into different staff types, while Panel C depicts the same for a metapopulation model where nurses were assigned to a strict subpopulation of patients. Panel D depicts the difference in proportional changes between the Metapopulation and Nurse-MD models.

## Metapopulation interactions

Evaluating the relationship between gamma ($\gamma$), the proportion of time a nurse interacts with their originally assigned patient group, and MRSA acquisitions was identified as non-linear (Fig 4), with progressively higher values of $\gamma$ resulting in drastically reduced rates of MRSA acquisitions. The segmented Poisson regression model identified a single change point, gamma* ($\gamma^*$), at 0.40 (95% Confidence Interval: 0.37, 0.42). This value reflects a nurse spending a slim majority of their time (40%) with the fifteen patients not directly assigned to them, with the remainder focused on the three patients who are. Values below $\gamma^*$ were well approximated by the Nurse-MD model, and values above it rapidly approached the stricter assignment of the Metapopulation model.

## Discussion

The reduction and control of healthcare-associated infections in recent decades has been a major accomplishment. In 2015, a point-prevalence survey found that HAIs affected roughly 3.2% of hospitalized patients in the United States, down from about 4% in 2011 [48].

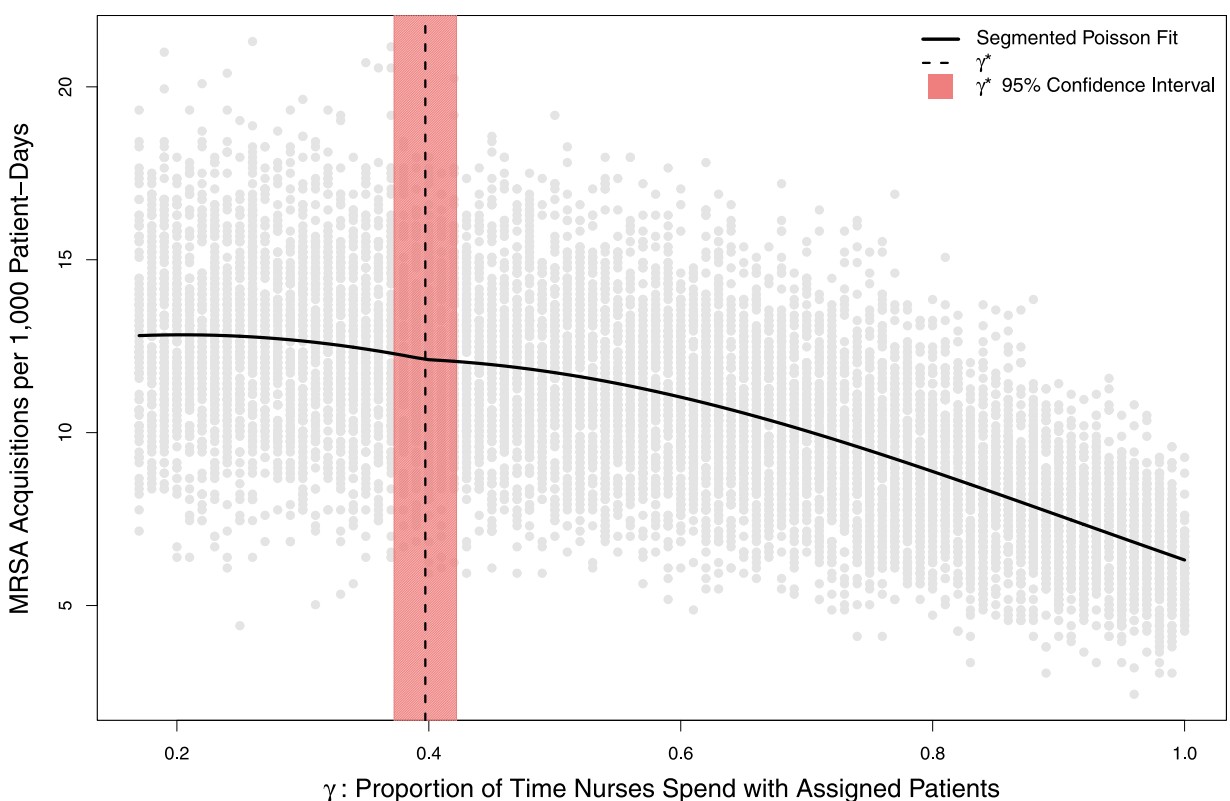

**Fig 4. Relationship between the proportion of time nurses spend treating patients outside their assigned group (γ) and cumulative MRSA acquisitions over 10,000 simulations, randomly sampling γ from a uniform distribution between 1/6 and 1.** Grey dots show an individual simulation, while the black line shows a segmented Poisson regression fit with linear and quadratic terms for γ. The vertical dashed line depicts the single segmentation point, γ*, to the left of which these more complicated models are adequately approximated by the Nurse-MD model where random mixing occurs. The shaded area shows the corresponding confidence interval.

Unfortunately, many drug-resistant or multidrug-resistant organisms (MDROs) remain an urgent or serious threat according to The Centers for Disease Control and Prevention (CDC), and certain pathogens, such as MRSA, continue to cause over 10,000 deaths annually [49]. The CDC continues to consider MRSA a serious threat to patient safety, and recent trends show that the incidence of hospital-onset bloodstream infections are no longer declining as observed for much of the last decade [50].

The evidence base for interventions to successfully address MRSA is mixed. For example, it has been difficult to quantify the effectiveness of MRSA screening and contact precautions, which has led to disagreement over their benefit [51–54]. However, more recent evidence clearly suggests that contact precautions among MRSA patients in Veterans Affairs acute care hospitals has a large reduction in transmission [55]. Other efforts, such as improved hand hygiene are often successful [56,57], but reducing MRSA acquisitions by means of improved transmission prevention continues to be a focus for hospital infection control efforts.

Even fairly subtle changes in model structure can greatly impact the estimated effectiveness of interventions. While remaining in the compartmental modeling framework, the more complex Metapopulation model was considerably more conservative when producing estimates of intervention effectiveness. While the models considered in this study had similar parameter sensitivity in terms of the *direction* of changes, the more highly structured models were relatively less sensitive. In all cases, the contact rate (ρ), probability of patient colonization (ψ), and

hand-decontamination ($\iota$) parameters had the largest impact, consistent with many of the known drivers of infection rates within hospitals. A model that assumes completely random mixing allows a higher degree of interaction with the patients and healthcare workers, resulting in an over-estimation of both the overall rate of MRSA acquisition and an over-estimation of the impact of interventions.

The sensitivity analyses also provided insight into the importance of accurately modeling different classes of healthcare worker. Our results suggest that the physician contact rate ($\rho_D$) had a very small effect on the change in cumulative MRSA acquisitions. Our model suggests that further exploration of nurse interaction and contact is most likely a more important focus for future infection control efforts.

The impact of changing the structure of a simulation ICU was, importantly, non-linear. The much simpler Nurse-MD model well-approximated the relatively more complex Metapopulation model when nurses spent between 16.7% (equivalent to random mixing) and 40% of their time with their assigned patients ($\gamma$). Informal estimates from two tertiary care academic medical centers in the southeastern United States estimated their ICUs at 80% and 90% (personal communication), an area of the parameter space where the impact of how one chooses to model the structure of an ICU has a pronounced impact.

## Conclusion

When combined, these results suggest that while compartmental models assuming random mixing and those that have more specified population interactions may give qualitatively the same answer as to the benefit of an intervention, the magnitude of these estimates may vary considerably, which has implications for cost-effectiveness models and other studies that rely on these estimates. Additionally, if the interventions suggested by the model are implemented in practice, the performance of the intervention may differ from the model's predictions due to the choice of how the population interacts. Finally, these results show that fitted parameters can vary considerably even for very similar models, suggesting even mild changes in model form necessitate refitting, and parameter estimates are not transportable from one model to another model with a differing population interaction structure.

The results of this study do have clinical implications. In an ICU setting, it is foreseeable that events like emergencies, breaks, or cross-coverage of nurses will occur with a reasonable degree of frequency. The COVID-19 pandemic's impact on staffing, the creation of COVID-19 specific wards in spaces not intended for critical care, and the demands of treating COVID-19 patients is a particularly vivid example of this [58]. Our model suggests that even relatively small increases in the rate at which these interactions occur can have outsized impacts on MRSA acquisition rates, which is likely to be true for other healthcare-associated pathogens as well.

Methodologically, our study suggests that the appropriateness of using simplified model structures for healthcare-associated infections is highly context dependent. In circumstances where a simple qualitative answer or generalized ranking is needed, a simplified compartmental model that does not attempt to represent patient-provider mixing patterns may be sufficient. The COVID-19 pandemic, which has demanded rapid turn-around of modeling results in a largely distributed fashion is an example of a circumstance where the tradeoff between speed and accuracy in model development is likely important.

Our results also suggest that ICUs with highly interactive nurse-patient populations may not need more complex models–and while this is likely not the case in developed world urban tertiary care academic medical centers, it may be true in other environments such as rural hospitals or low and middle income country settings. Which model structure is most appropriate

in these settings is empirically measureable, with an estimate of gamma informing both model structure and parameterization.

This study has several limitations. While the Metapopulation model is a more granular representation of a hospital population than the more-common SST model, it too is a simplification. Similarly, the parameter estimates used in the model are imperfect. It is likely that the hand hygiene rate is likely higher than the rates occurring in many hospitals, as reported rates are often substantially inflated. However, these estimates are drawn primarily from the established literature, and represent the field's best understanding of the underlying processes.

Other limitations include the structure of the model–it focuses specifically on healthcare worker and patient interactions and does not account for interactions with individuals other than nurses and the physician. For example, interactions among patients, visitation by family and friends, medical or radiological technicians performing a specific procedure, etc. are not represented. Some of these individuals, such as technicians, arguably add an additional random element that would connect otherwise partially or wholly separated patients, and correspondingly increase infection rates. Others, such as visitors, primarily represent risk to a single patient. Similarly, transmission purely through environmental contamination is not represented. These simplifications were primarily chosen to make the illustration of the impact and potential necessity of moving to a more complex model structure as clear as possible.

This study shows that the random mixing assumption results in an over-estimation of both the overall rate of MRSA acquisition and an over-estimation of the impact of interventions as expressed as changes in model parameter values. In many–but not all–circumstances the use of more complex models is likely warranted, even for small-scale models of a single ICU. Importantly however, this necessity can be established not using a heuristic or qualitative assessment based on a modeler's bias or preferences, but with a straightforward parameter estimate. The estimation of this parameter in several different healthcare contexts, including rural and LMIC settings, is ongoing. More broadly, the results of this analysis show the need for structural sensitivity analysis to accompany analysis of parameter uncertainty. While all the models explored here are adaptations of familiar compartmental forms, and the steps between them relatively simple, they can have dramatic impacts on the results of the models–impacts that can be amplified when these results are incorporated into larger models, cost-effectiveness analyses, guidelines and position papers, etc.

## Supporting information

**S1 Table. Transitions and equations for the Single Staff Type (SST) Model of MRSA Acquisition.**
(DOCX)

**S2 Table. Transitions and equations for the Nurse-MD Model of MRSA Acquisition.**
(DOCX)

**S3 Table. Transitions and equations for the Metapopulation Model of MRSA Acquisition.**
(DOCX)

## Acknowledgments

The authors would like to acknowledge Justin O'Hagan for his thoughtful input.

MinD Healthcare Members: Eric T. Lofgren, PhD, Washington State University; Eili Klein MA, PhD, Johns Hopkins School of Medicine; Sarah Rhea DVM, MPH, PhD, RTI International; Matthew Samore, MD, University of Utah School of Medicine; Alberto Segre, MS, PhD, University of Iowa.

## Author Contributions

**Conceptualization:** Matthew S. Mietchen, Eric T. Lofgren.

**Data curation:** Matthew S. Mietchen, Eric T. Lofgren.

**Formal analysis:** Matthew S. Mietchen, Christopher T. Short, Eric T. Lofgren.

**Funding acquisition:** Eric T. Lofgren.

**Investigation:** Matthew S. Mietchen, Eric T. Lofgren.

**Methodology:** Matthew S. Mietchen, Christopher T. Short, Matthew Samore, Eric T. Lofgren.

**Project administration:** Eric T. Lofgren.

**Resources:** Eric T. Lofgren.

**Software:** Matthew S. Mietchen, Eric T. Lofgren.

**Supervision:** Eric T. Lofgren.

**Validation:** Christopher T. Short, Eric T. Lofgren.

**Visualization:** Matthew S. Mietchen, Eric T. Lofgren.

**Writing – original draft:** Matthew S. Mietchen, Eric T. Lofgren.

**Writing – review & editing:** Matthew S. Mietchen, Christopher T. Short, Matthew Samore, Eric T. Lofgren.

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
