## [Decision Letter · Decision Letter 0]

13 Sep 2019

Dear Dr Lofgren, 

Thank you very much for submitting your manuscript 'Population Structure Drives Differential Methicillin-resistant Staphylococcus aureus Colonization Dynamics' for review by PLOS Computational Biology. Your manuscript has been fully evaluated by the PLOS Computational Biology editorial team and in this case also by independent peer reviewers. The reviewers appreciated the attention to an important topic, but they raised substantial concerns about the paper. Based on the reviews and editorial discussions, we regret that we will not be able to accept this manuscript for publication in the journal. 

The reviews are copied below, and we hope they may help you should you decide to revise the manuscript for submission elsewhere. We are sorry that we cannot be more positive on this occasion, but hope that you appreciate the reasons for this decision and that you will consider PLOS Computational Biology for other submissions in the future. 

Thank you again for your support of PLOS Computational Biology and open-access publishing. Please do not hesitate to get in touch (via ploscompbiol@plos.org) if we can provide any further assistance. 

Sincerely, 

Benjamin Althouse

Associate Editor

PLOS Computational Biology 

Rob De Boer

Deputy Editor

PLOS Computational Biology 

[LINK]

I agree with reviewer 1 on the lack of novelty of this manuscript and it missing mention to much previous work on this topic.

Reviewer's Responses to Questions

**Comments to the Authors: **

Reviewer #1: In the paper by Mietchen et al, the authors build upon a stochastic compartmental model to examine MRSA transmission in an ICU. They compare and contrast random vs. non-random mixing approaches in patient care and conclude that random mixing assumptions may be inappropriate. I congratulate the authors on developing a computational model examining an important topic (which I recognize takes many hours to do) yet I find their conclusion somewhat underwhelming as, in fact, this has been a known limitation in the epidemiology modeling field for some time. We have seen the rise of approaches that relax or omit this assumption over the past decade, but these other approaches are not mentioned in the manuscript. This ties in with my main concern: the background and discussion do not acknowledge the wealth of work done in modeling infectious diseases that do not assume a random mixing approach. I also have the following comments for the authors to consider to additionally improve this work.

Introduction: 

• Please define consistent care teams: Are the authors referring to same providers, roles, or something else? What about shift changes?

• “grouping particular types of patients together”: Please use the term “cohorting”. This is the standard term in hospital infection prevention. 

• The lit review seems to have missed some recent papers that examine how the built hospital environment and patient care networks impact MRSA transmission in the ICU. Some of these papers are based in the neonatal ICU, but have modeling implications for the adult ICU as well. 

• Re: violating the independence assumption of many statistical techniques. I agree with the authors, and thus researchers use a variety of models with correlated errors to account for this occurrence. Would acknowledge that there are techniques that do not depend on the independence assumption.

• Re: limited or no control groups. But what about multicenter studies that either do a site-level randomization, or purely observational work? While differences between sites exist, the process of delivering care in an ICU is fairly standardized in the U.S. 

• The authors state that “widely used models generally assume random mixing between healthcare workers and patients,” but there are no cites to back this claim up. This may have been true a decade or so ago, but use of network models relax the random mixing approach and in fact are now expected and move beyond compartmental modeling approaches. This is why in the epidemiology literature we see the increase in ABMs and network models, yet there is no acknowledgement of these in the paper. Examples exist in both the healthcare and non-healthcare environments.

Methods:

• Why include the SST model? This is not realistic. In fact, the majority of care in an ICU is provided by the nurse and not the physicians. Hand hygiene can also be differentially modeled by provider role.

• More detail regarding the setting is needed (brought in from the referenced publication). Is this meant to reflect an actual ICU or a hypothetical ICU? What type of ICU? How many admissions per year? Especially since the authors rely on parameter estimates from [17] that combines med/surg ICUs, these could be very different in terms of risk for organism colonization and infection.

• Re: MRSA colonization/detection assumption. Does this ICU carry out some kind of surveillance for MRSA colonization? Further, if an individual is colonized with MRSA, is a decolonization regimen performed (aside from the “natural” decolonization employed)? These are important considerations to accurately model MRSA transmission.

• Perhaps the authors can consider stratifying potential for contamination/transmission based on types of interactions with the patients. For example, there are a lot of routine, more mundane activities that confer lower risk, and fewer, invasive activities that confer higher risk. This would add an interesting and novel aspect to this work.

• While the model [appears to be] calibrated, was it also validated to ensure it reflects some kind of real ICU? This will make the findings more relevant for infection preventionists. Especially because in the Discussion the authors state that these models are useful “only if they can represent the population and transmission dynamics of a hospital.” This goes back to my earlier comment to better describe the setting.

• I applaud the authors for releasing their analytic code and data. More research groups should do this.

Results/Discussion:

• Figure 4/number of MRSA acquisitions: Please include a denominator, such as patient-days, when presenting incidence. This will make the findings comparable.

• Would be nice to include estimates of the proportion of MRSA transmissions due to the provider role under each scenario.

• Can the authors provide an intuitive interpretation of gamma=0.4? How does this finding potentially impact patient care dynamics in an ICU?

• The discussion needs to be expounded to compare and contrast to other work in this area, as well as provide recommendations for how the findings are useful for infection control in an ICU setting. 

• The authors state that the “finding has broad implications for staffing levels and hospital policy." Such as ???

• The authors note the lack of non-nurse/MD personnel and visitors in their model. How does this omission potentially impact findings?

• How does the MRSA colonization/detection assumption impact findings? Especially because this is an important assumption and not reflective of how MRSA is actually detected. Suggest the authors bring in a surveillance aspect to this, and (possibly) decolonization if they use this in their ICU. 

• Patient cohorting is mentioned in the introduction, but not returned to in the Discussion or elsewhere. Another opportunity to add an interesting angle to this work would be to compare and contrast cohorting and isolation of MRSA carriers.

Reviewer #2: This paper uses a modeling-based approach to compare several different structures of hospital ICU population interactions and how that affects MRSA acquisition rates. The goal seems to be to inform future models of MRSA acquisition within ICU settings, noting that simpler models overestimate both MRSA acquisition and the potential effects of interventions. This seems important and the importance of the findings could be highlighted more in the introduction.

The science in this work appears very well thought out, but a few improvements in the writing could make it easier for the reader to follow and understand earlier on in the paper.

1) short title includes “ICU”, main title should include “ICU” too

2) the term “Population structure” doesn’t seem quite right—perhaps “population interaction structure” is more accurate?

3) whenever feasible, it would be helpful for the reader to write out what the parameter is measuring, rather than just the parameter letter

4) it would be helpful to have a more general description of what the models measure and what they take into account earlier on (rather than just the population interaction structures)

5) it would also be helpful to label whatever parameters you could on the diagrams

6) at line 306, is the word contact correct? "The altered contact patterns in the Meta-population model thus needs substantially higher per-contact colonization probabilities to sustain the same level of contact.”

7) Figure 5: it would be easier to compare the 4 panels if the parameters were either in the same order or color-coded

**Have all data underlying the figures and results presented in the manuscript been provided?**

Reviewer #1: Yes

Reviewer #2: Yes

PLOS authors have the option to publish the peer review history of their article (what does this mean?). If published, this will include your full peer review and any attached files.

Reviewer #1: No

Reviewer #2: No

---

## [Decision Letter · Decision Letter 1]

9 Dec 2021

Dear Dr. Lofgren,

Thank you very much for submitting your manuscript "Examining the Impact of ICU Population Interaction Structure on Modeled Colonization Dynamics of Staphylococcus aureus" for consideration at PLOS Computational Biology.

As with all papers reviewed by the journal, your manuscript was reviewed by members of the editorial board and by several independent reviewers. In light of the reviews (below this email), we would like to invite the resubmission of a significantly-revised version that takes into account the reviewers' comments.

We cannot make any decision about publication until we have seen the revised manuscript and your response to the reviewers' comments. Your revised manuscript is also likely to be sent to reviewers for further evaluation.

Sincerely,

Benjamin Althouse

Associate Editor

PLOS Computational Biology

Rob De Boer

Deputy Editor

PLOS Computational Biology

Reviewer's Responses to Questions

**Comments to the Authors:**

Reviewer #2: On re-review of this manuscript by Mietchen et al, I do find that I understand it better, but largely because of their responses to reviewers. More of these responses should make it into the paper.

This is good work that will help inform when certain models are appropriate to use, but the writing needs a lot of work. The introduction and discussion sections especially need to be more focused and set up the question better. And relevant models, such as from the 5 CDC sites, need to be cited. Much more citation of the modeling literature is needed—especially in the discussion section, which currently has zero citations.

General

• The main point of sentences should more often be at the beginning of the sentence so it’s easier to follow.

• For future versions, please use track changes correctly.

o It appears that you just cut everything and pasted—making it appear that everything was entirely re-written, which is not the case (the 1st sentence of the intro is the same.

• The writing throughout could be much more concise.

Abstract

• The “background” section isn’t actually background or justification for why you’re comparing these models—you need to set up your story. The author summary did a better job of this.

Introduction

• The introduction currently focuses too much and too early on the specifics of MRSA and infection prevention. It needs to highlight much more and earlier on, the importance of models in healthcare settings, the desire for simple models, and the question of whether simple models are appropriate.

• The specific details of MRSA aren’t as important and can be briefly summarized and cited.

Methods

• Model Structure section is too wordy. Can be shortened substantially.

• Cite Table 1 earlier on in your description of the model structures, and also cite the supplement with the equations.

Figures

• Note in all figure legends that arrows represent parameters found in table 1, and that equations are found in supplement.

**Have the authors made all data and (if applicable) computational code underlying the findings in their manuscript fully available?**

Reviewer #2: Yes

PLOS authors have the option to publish the peer review history of their article (what does this mean?). If published, this will include your full peer review and any attached files.

Reviewer #2: No
---

## [Decision Letter · Decision Letter 2]

19 Apr 2022

Dear Dr. Lofgren,

Thank you very much for submitting your manuscript "Examining the Impact of ICU Population Interaction Structure on Modeled Colonization Dynamics of Staphylococcus aureus" for consideration at PLOS Computational Biology. As with all papers reviewed by the journal, your manuscript was reviewed by members of the editorial board and by several independent reviewers. The reviewers appreciated the attention to an important topic. Based on the reviews, we are likely to accept this manuscript for publication, providing that you modify the manuscript according to the review recommendations.

Sincerely,

Benjamin Althouse

Associate Editor

PLOS Computational Biology

Rob De Boer

Deputy Editor

PLOS Computational Biology

[LINK]

Reviewer's Responses to Questions

**Comments to the Authors:**

Reviewer #2: Please see attached review.

**Have the authors made all data and (if applicable) computational code underlying the findings in their manuscript fully available?**

Reviewer #2: Yes

PLOS authors have the option to publish the peer review history of their article (what does this mean?). If published, this will include your full peer review and any attached files.

Reviewer #2: No

Figure Files:

Data Requirements:

Reproducibility:

References:

---

## [Editor Report · Decision Letter 3]

3 Jul 2022

Dear Dr. Lofgren,

We are pleased to inform you that your manuscript 'Examining the Impact of ICU Population Interaction Structure on Modeled Colonization Dynamics of Staphylococcus aureus' has been provisionally accepted for publication in PLOS Computational Biology.

Best regards,

Rob J. De Boer

Deputy Editor

PLOS Computational Biology

Rob De Boer

Deputy Editor

PLOS Computational Biology

---

## [Editor Report · Acceptance letter]

20 Jul 2022

PCOMPBIOL-D-19-01296R3 

Examining the Impact of ICU Population Interaction Structure on Modeled Colonization Dynamics of *Staphylococcus aureus*

Dear Dr Lofgren,

I am pleased to inform you that your manuscript has been formally accepted for publication in PLOS Computational Biology. Your manuscript is now with our production department and you will be notified of the publication date in due course.

With kind regards,

Olena Szabo
